# Personalized Treatments Based on Laryngopharyngeal Reflux Patient Profiles: A Narrative Review

**DOI:** 10.3390/jpm13111567

**Published:** 2023-10-31

**Authors:** Jerome R. Lechien

**Affiliations:** 1Division of Laryngology and Broncho-Esophagology, Department of Otolaryngology-Head Neck Surgery, EpiCURA Hospital, UMONS Research Institute for Health Sciences and Technology, University of Mons (UMons), B7000 Baudour, Belgium; jerome.lechien@umons.ac.be; 2Phonetics and Phonology Laboratory (UMR 7018 CNRS, Université Sorbonne Nouvelle/Paris 3), Department of Otorhinolaryngology and Head and Neck Surgery, Foch Hospital, School of Medicine, (Paris Saclay University), 92150 Paris, France; 3Department of Otorhinolaryngology and Head and Neck Surgery, CHU Saint-Pierre, School of Medicine, B1000 Brussels, Belgium; 4Research Committee of the Young Otolaryngologists of the International Federation of Otorhinolaryngological Societies (YO-IFOS), 92150 Paris, France; 5Department of Otolaryngology, Elsan Hospital, 92150 Paris, France

**Keywords:** laryngopharyngeal, gastroesophageal, reflux, otolaryngology, head neck surgery, voice, personalized, laryngeal, future, treatment, therapy

## Abstract

Objective: To review the current findings of the literature on the existence of several profiles of laryngopharyngeal reflux (LPR) patients and to propose personalized diagnostic and therapeutic approaches. Methods: A state-of-the art review of the literature was conducted using the PubMED, Scopus, and Cochrane Library databases. The information related to epidemiology, demographics, clinical presentations, diagnostic approaches, and therapeutic responses were extracted to identify outcomes that may influence the clinical and therapeutic courses of LPR. Results: The clinical presentation and therapeutic courses of LPR may be influenced by gender, age, weight, comorbidities, dietary habits and culture, anxiety, stress, and saliva enzyme profile. The clinical expression of reflux, including laryngopharyngeal, respiratory, nasal, and eye symptoms, and the hypopharyngeal–esophageal multichannel intraluminal impedance-pH monitoring profile of patients are important issues to improve in patient management. The use of more personalized therapeutic strategies appears to be associated with better symptom relief and cures over the long-term. The role of pepsin in LPR physiology is well-established but the lack of information about the role of other gastrointestinal enzymes in the development of LPR-related mucosa inflammation limits the development of future enzyme-based personalized diagnostic and therapeutic approaches. Conclusion: Laryngopharyngeal reflux is a challenging ear, nose, and throat condition associated with poor therapeutic responses and a long-term burden in Western countries. Artificial intelligence should be used for developing personalized therapeutic strategies based on patient features.

## 1. Introduction

Laryngopharyngeal reflux (LPR), also called respiratory reflux, may be defined as an inflammatory condition of the upper aerodigestive tract tissues that is related to the direct and indirect effects of gastroduodenal content reflux, which induces morphological changes in the upper aerodigestive tract [1]. The pathophysiology of LPR has not been fully elucidated, while the contributing factors remain unclear [2,3]. The clinical presentation is characterized by non-specific symptoms and findings, which makes the clinical diagnosis challenging [4,5]. Hypopharyngeal–esophageal multichannel intraluminal impedance-pH monitoring (HEMII-pH) may be considered as the gold standard for the diagnosis [1]. However, HEMII-pH is poorly available in many hospitals, expensive, and inconvenient for patients [6,7]. The non-specificity of symptoms and findings leads to under- or over-estimation of LPR [8] and patients are often unnecessarily treated with antiacid therapeutics (e.g., proton pump inhibitors (PPIs)) [9]. Indeed, LPR treatment is mainly based on PPIs that were never demonstrated to be superior to a placebo [9,10]. As of 2023, LPR remains a controversial and challenging condition, affecting 10% to 30% of outpatients consulting otolaryngology offices [1]. An increasing number of studies suggested the existence of several LPR profiles [11,12,13,14,15], which should be considered for more personalized diagnostic and therapeutic approaches.

In the present review, the findings of the literature on the existence of several profiles of LPR were reviewed and discussed to propose personalized diagnostic and therapeutic approaches.

## 2. Physiology

### 2.1. Gastroduodenal Enzymes

The backflow of gastroduodenal content into the upper aerodigestive tract mucosa mainly occurs in the daytime and when they are upright occurs through weakly acidic or alkaline pharyngeal reflux events [1,13]. The pH of the reflux event may increase from the distal to the proximal esophagus. The mechanisms underlying this increase in pH remain unknown and should involve the secretion of bicarbonate into the esophageal mucosa [13]. Nowadays, basic science and clinical studies support a key role of pepsin, which is associated with intra- and extracellular injuries of the upper aerodigestive tract mucosa [2,3]. The deposition of pepsin in laryngopharyngeal tissues leads to significant macroscopic and microscopic histopathologic changes including epithelial cell dehiscence, microtraumas, inflammatory infiltrates, mucosal drying, and epithelial thickening [16]. In the case of an extracellular alkaline environment, the internalization of pepsin into the acidic environment of the Golgi apparatus may reactivate the pepsin, leading to intracellular damage and cell apoptosis [2,3]. A few years ago, Sereg-Bahar et al. and De Corso et al. reported that LPR patients may have a significantly higher bile salt concentration in their saliva compared to asymptomatic individuals [17,18]. Interestingly, the saliva bile salt concentration was predictive of the severity of LPR [18]. The toxicity of bile salts on laryngopharyngeal tissue was confirmed in experimental studies that supported a potential carcinogenic role of bile acids in laryngeal squamous cell carcinoma [19]. According to the literature, pepsin and bile acids may be considered important etiological factors for upper aerodigestive tract mucosa inflammation and related symptoms and findings. Because pepsin is active in acidic or weakly acidic pHs, ranging from 1.5 to 6.0 [20], while conjugated or deconjugated bile acids may be activated in a large range of pHs (acidic, weakly acidic, or alkaline), the consideration of saliva concentrations of pepsin and bile salts in patient may lead to more personalized treatments by combining PPIs, magaldrate, or alginate. However, a personalized approach based on enzyme profiles requires future studies to determine the role of all enzymes in the development of mucosal lesions and inflammation.

### 2.2. Diet and Lifestyle Habits

The diet has a significant impact on gastroesophageal motility and esophageal sphincter tonicity [21]. Clinical and basic science studies showed that fatty foods decrease the lower esophageal sphincter (LES) pressure and increase the esophageal acid exposure [22,23,24,25,26]. The refluxogenic potential of fat is due to its slow gastric emptying time, which is related to lipid digestion and the higher number of transient relaxations of the LES [27]. Similarly, chocolate syrup was associated with a decrease in LES pressure and a greater esophageal acid exposure time [25,28]. A decrease in LES was similarly observed in patients who consumed carbonated beverages [29], beverages with caffeine [30], or alcohol [31,32,33,34,35]. Concerning theine and tea, a recent meta-analysis supported that there are several subgroups of tea drinkers, with some developing reflux, and others not [36]. The same reasoning was suggested for coffee and caffeine; however, there are inter-individual differences in caffeine metabolisms, leading to controversial results [37,38]. Fried and spicy foods, such as chili, did not impair the overall esophageal motility but they did affect the LES tonicity [39,40]. Raw vegetables may be considered refluxogenic foods due to their slower gastric emptying time associated with the fiber digestion [21]. Tomato-based products increased the risk of gastroesophageal reflux disease (GERD) through the two prominent organic acids found in tomato (citric and malic acids), which are triggers of acid reflux in prone individuals and higher tomato consumers [41,42]. Similarly, onions may increase the number of reflux episodes and the esophageal acid exposure [43], while mint should relax the LES in some patients [44,45]. To date, practitioners advise a standardized alkaline, low-fat, low-high released sugar, high-protein diet because it is associated with greater symptom improvement in LPR patients treated with PPIs compared to those treated with PPIs alone [5,20]. However, the impact of diet on symptom relief probably depends on the baseline diet of patients, and its contribution to the occurrence of pharyngeal reflux events. Some patients with a high refluxogenic diet and few other contributing factors, such as autonomic nerve dysfunction or tobacco consumption, will probably see a significant impact of an antireflux diet, while those with a low refluxogenic diet at baseline may have poor benefits with diet intensification. In the same vein, because most pharyngeal reflux events occur in the daytime and when upright, the importance of the elevation of the head of the bed and avoidance of meals before sleep was not demonstrated in LPR. The importance of diet was supported in a recent study where 54% of LPR patients treated with only through a dietary intervention experienced relief of symptoms [46]. More personalized diet recommendations should improve the management of patients but it needs to consider the patient’s baseline diet. A list of recommended foods and beverages and those to avoid is available in Appendix A for clinical use.

### 2.3. Autonomic Nerve Dysfunction

The vagus nerve has key roles in laryngopharyngeal mucosa sensitivity, esophageal sphincter tonicity, and gastroesophageal motility. The balance between the para-sympathetic and sympathetic systems is important in digestive homeostasis and may be impaired when the sympathetic system is abnormally activated. Stress, anxiety, and depression are all conditions that may activate the sympathetic system and, consequently, decrease the para-sympathetic activity of the vagus nerve. The assessment of high and low frequencies of heart rate variability is an objective approach to measure the sympathetic and para-sympathetic activities [47]. Huang et al. reported that suspected LPR patients had anxiety, significantly lower high frequency heart rate variability, and a higher low frequency/high frequency ratio than asymptomatic individuals, which demonstrated poor autonomic modulation and higher sympathetic activity. The imbalance between the sympathetic and para-sympathetic systems was confirmed by Wang et al. who similarly reported higher low frequency/high frequency ratios in suspected LPR patients compared to controls [48]. In 2017, Joo et al. found that LPR was more prevalent in patients with depression than in those without (45.6% vs. 27.0%), which supported the role of autonomic nerve dysfunction in LPR disease [49]. From a therapeutic standpoint, Heading et al. demonstrated that depression and anxiety negatively influenced the therapeutic response of patients [50]. However, to date, studies objectively investigating the relationship between autonomic nerve dysfunction and hypopharyngeal reflux events using HEMII-pH are lacking.

The severity of reflux based on HEMII-pH, anatomical and physiological gastroesophageal impairments, autonomic nerve dysfunction, and diet features are all outcomes that may influence the evolution of LPR over time. Thus, based on reflux symptoms, the clinical course of LPR patients may be categorized into acute, recurrent, or chronic [51].

The key physiological features can be summarized as follows:-Laryngopharyngeal symptoms and findings may be due to the inflammation of the mucosa, which is related to pepsin and/or bile acidic toxicity.-Pepsin is mainly active in acidic or weakly acidic environment, whereas bile acids may be active in acidic, weakly acidic or alkaline environment.-The consideration of pepsin and bile salt saliva concentrations should indicate a more personalized treatment, combining antiacids and over-the-counter drugs.-The composition of foods and beverages may influence the gastroesophageal motility and sphincter tonicity and, consequently, the occurrence of pharyngeal reflux events and deposition of enzymes into the upper aerodigestive tract mucosa.-Depression, anxiety, stress, and the related autonomic nerve dysfunction are more commonly found in patients with symptoms and findings of LPR, which suggests a key role of the autonomic nervous system in the physiology of LPR.-The patients’ baseline diet, personality, lifestyle, and potential triggers of autonomic nerve dysfunction need to be considered to propose a more personalized short-to-long-term treatment without medication as much as possible.

## 3. Patient Features

The physiology of the human body is influenced by several factors including gender, age, weight, comorbidities, and medications. The consideration of these factors is known to improve the overall management of many diseases [52,53]. Many studies have investigated the influence of gender, age, or overweight on the clinical presentation and therapeutic response of LPR.

### 3.1. Gender

Females and males differ in anatomical, hormonal, and chromosomal features. Several genes involved in inflammation are located on the X chromosome and can escape X chromosome inhibition [54]. Consequently, females have a higher potential for inflammation compared to males, which leads to a higher severity of symptoms in many respiratory diseases, such as asthma [55] or chronic rhinosinusitis [56]. In LPR, females commonly present with higher baseline symptom scores, such as RSS, compared to males [57]. Moreover, it has been showed that females may require more time to reach symptom relief throughout treatment compared to males. This is particularly observed in laryngology because females with suspected LPR may be more at risk for LPR-related dysphonia than males [58,59,60].

### 3.2. Age

Most clinical studies supported that elderly LPR patients have lower symptom scores and related quality of life impact at baseline compared to younger patients [57,61]. Practitioners may consider a longer duration of treatment in elderly individuals compared to younger patients (6 months vs. 3 months) because their symptoms need more time to disappear [57]. The pattern of symptoms may also vary according to age. Due to neurologic deterioration of the terminal-sensitive nerve endings, elderly individuals with LPR or GERD may not experience the typical set of LPR-related symptoms [62]. From a therapeutic standpoint, Lee et al. reported that elderly patients were more likely to not respond to PPI therapy than younger patients [63], which supports our recent findings [57] suggesting that the therapeutic period for elderly LPR patients needs to be longer compared to younger patients. The differences in mucosa healing and symptom relief between young and old patients may be due to age-related changes, such as hormonal secretion, nutritional status, mucosal biomolecular composition, vascular supply of tissue, and mucosal defense mechanisms [64].

### 3.3. Overweight

The negative impact of overweight and obesity on the esophageal sphincter tonicity, GERD complications, and severity of symptoms has been well-known for a long time [12,65]. In LPR, studies found that most LPR patients are not obese [12,65,66]. However, patients with overweight may report a higher prevalence of GERD and acid pharyngeal reflux events according to HEMII-pH compared to others, while their symptoms were more severe according to RSS [12]. Similar findings were found by Halum et al. who observed that LPR patients with an increased body mass index reported more frequently GERD in a dual-probe pH study [66]. From a pathophysiological standpoint, an increased number of transient LES relaxation episodes leads to an increased risk of distal-to-proximal reflux events, some of which reach the pharynx [12]. The higher rate of GERD and acid pharyngeal reflux events in overweight and obese patients may suggest the use of PPIs in addition to alginate as personalized empirical therapeutic approach.

### 3.4. Medical and Surgical Conditions

Some digestive or respiratory conditions have been suspected to be associated with the development of LPR or the recurrence of symptoms after an empirical therapeutic trial. Recently, Balouch et al. reported that gluten sensitivity may mimic or aggravate LPR disease [67]. In the same vein, the authors observed that a gluten-free diet may be advised to patients with recalcitrant LPR, especially if blood test abnormalities suggest a gluten sensitivity. This diet may be associated with a significant improvement in laryngeal findings after 3 months of diet [67]. Lactose intolerance was suggested as an additional condition that may mimic or aggravate LPR [68] but only pediatric studies confirmed this association in GERD children [69]. Sleeve gastrectomy is another theoretical contributing factor for LPR [68]. A recent study showed that patients who underwent sleeve gastrectomy may develop de novo GERD symptoms in 16.1% of cases, while this proportion increased for elderly patients and lower body mass index patients [70]. Other LPR-associated conditions may include histamine sensitivity [71], cholecystectomy [68], inlet patch [72], ineffective esophageal peristalsis [73], gastroparesis [68], and rumination or aerophagia [68,74,75].

Because LPR is associated with non-specific symptoms and findings, many ear, nose, and throat conditions may mimic LPR symptoms in patients without pharyngeal reflux events according to HEMII-pH. These conditions need to be excluded at the time of diagnosis or in patients with recalcitrant symptoms, which may be difficult in departments without HEMII-pH devices. Among them, alimentary food intolerance, infectious (e.g., Chlamydia Pneumoniae or Mycoplasma Pneumoniae), toxin- (e.g., tobacco, vaping) or drug-induced (inhaled corticosteroids) pharyngolaryngitis, may mimic LPR symptoms [68]. Nowadays, all of these conditions are considered differential diagnoses of LPR, but they could also favor the development of LPR through the reduction in mucosa defense mechanisms or the increase in mucosa inflammation and sensitivity to pepsin aggression. Future studies are needed to investigate the impact of these conditions on laryngopharyngeal cells and the related mucosa weakening. Most laryngeal and extra-laryngeal symptoms and findings associated with LPR are described in the reflux symptom score (Figure 1) [76] and reflux sign assessment (Figure 2) [77].

The key clinical findings can be summarized as follows:-The severity of laryngopharyngeal symptoms and findings may be influenced by the age, gender, or body mass index of the patient.-Elderly and female patients may require more time to see symptom relief because the symptoms will continue to improve from until 6 months posttreatment, while the symptoms of responder males commonly improve by 3-months posttreatment.-Elderly patients may report lower baseline LPR and GERD symptom scores than younger patients, while they may have silent esophageal complications of GERD.-Some conditions may favor the development of LPR or recalcitrant symptoms and findings, including gluten sensitivity, lactose intolerance, or histamine sensitivity. These conditions need to be considered for the duration of treatment and throughout the follow-up of patients.-According to the IFOS classification [51], LPR may present as acute, recurrent, or chronic disease. To date, the influence of age, gender, overweight/obesity, or other contributing factors remains unknown. The identification of epidemiological factors contributing to both the recurrence of symptoms or the chronic course of the disease makes sense regarding the cost burden of LPR in Western populations [78].

## 4. Additional Examination Features

### 4.1. The Impedance-pH Monitoring Profile

According to the Dubai Consensus criteria for the definition and diagnosis of LPR (Table 1) [79], the LPR diagnosis is confirmed with the identification of >1 pharyngeal reflux events using HEMII-pH. In addition to its diagnostic role, HEMII-pH provides an LPR profile, including information on the time (daytime, nighttime, post-meal), the position (upright, supine), the pH (acidic, weakly acidic, alkaline), the composition of events (gaseous, liquid, mixed), as well as the coexistence of GERD [14,80]. Initially, Muderris et al. observed that 24% of LPR patients had pharyngeal reflux events but normal acid exposure in the low esophagus, whereas only 68% of proximal esophageal reflux events reached the hypopharynx [81]. This was corroborated by other studies that reported that patients do not experience GERD during HEMII-pH or at the gastrointestinal endoscopy in more than 50% of cases [14,80,82]. In most LPR cases, pharyngeal reflux events occur in the daytime and in upright positions (Figure 3). While the gaseous reflux event is often acidic in the low esophagus, studies mainly observed that LPR is composed of weakly acidic or alkaline pharyngeal reflux events, which is due to an increase in the pH from the distal to the proximal esophagus [14,80,82,83]. The mechanisms underlying this increase in pH remain unknown and could involve the secretion of bicarbonate into the esophageal mucosa. Interestingly, the analyses of the HEMII-pH tracing revealed that 74% of pharyngeal reflux events occurred outside the 1 h post meal time period, whereas 20% and 6% of events occurred during the 1 h post meal period and nighttime, respectively [80]. In this study, 58% of patients did not have nighttime reflux events [80]. However, we may observe reverse profiles using HEMII-pH for patients with only nighttime pharyngeal reflux events (Figure 3).

The analysis of the HEMII-pH tracing makes sense for the treatment. Indeed, according to the features of LPR at the HEMII-pH (pH, composition, and time), practitioners may prescribe PPI or not, post-meal/bedtime alginate or magaldrate [84].

### 4.2. Oropharyngeal and Nasopharyngeal pH Monitoring

Oropharyngeal pH monitoring (Restech Dx-pH monitoring) was developed for the diagnosis of LPR in ear, nose, and throat consultation [85]. However, to date, there is no agreement or international consensus guidelines for using oropharyngeal pH monitoring in daily practice with standardized criteria, which may be attributed to the low number of studies assessing normative data in asymptomatic individuals and the lack of information of esophageal events [79,86]. The lack of an esophageal sensor is the main controversial point of oropharyngeal pH monitoring because some authors argued that there should be false positive events regarding probe movements or mucosa dryness. In practice, this limitation of the lack of oropharyngeal pH monitoring was never demonstrated. In clinical practice, oropharyngeal pH monitoring could be an interesting diagnostic approach for the detection of nasopharyngeal reflux events. Indeed, the gastric content of pharyngeal reflux events may reach the sinonasal mucosa, Eustachian tube, and tears, which is associated with dry eyes, otitis media, Eustachian tube dysfunction, dry nasal mucosa, nasal crusts, or mulberry turbinate (Figure 4) [87,88,89,90]. Because there is no nasopharyngeal–hypopharyngeal–esophageal impedance-pH monitoring probe available in the market, the placement of oropharyngeal-pH monitoring sensors in the nasopharynx may be used to confirm nasopharyngeal reflux disease in such patients [91].

### 4.3. High-Resolution Manometry

To date, high-resolution manometry, GI endoscopy, and pepsin saliva concentration measurements are not evidence-based examinations indicated in all LPR patients [79]. However, according to the patient’s complaints or history, they may be useful in detecting digestive conditions associated with LPR development or therapeutic resistance. High-resolution manometry may detect/assess some anatomic and physiologic esophageal characteristics associated with LPR including esophageal sphincter tonicity and length, intrabolus pressure, proximal or distal esophageal body contractility, intra-abdominal esophagus length, and complete bolus clearance [92,93]. Van Daele reported that primary esophageal motility disorders, as per the Chicago classification, were detected in 43–63% of patients with LPR symptoms, including ineffective esophageal motility (31–41%), a hypercontractile esophagus (4–13%), and disorders of the esophagogastric junction outflow (8–9%) [94]. Borges et al. reported that some manometric features may be correlated with the severity of symptoms and reflux, such as the proportion of failed swallows [13,95]. However, these manometric findings are not specific to LPR, as they are also seen in GERD or other non-LPR conditions making the use of esophageal manometry in first-line management of LPR controversial. High-resolution manometry may be indicated in patients with treatment resistance or in individuals with LPR and a history of esophageal motility disorder [79].

### 4.4. Gastrointestinal Endoscopy

In the same vein, patients with a history of GERD complications, including esophagitis, esophageal hemorrhage, stricture, Barrett’s esophagus, and adenocarcinoma, may require transnasal or classical GI endoscopy [79,96]. Due to the reduction in GERD-symptom sensitivity with aging, elderly individuals may also benefit from GI endoscopy in baseline check-ups. Practitioners should keep in mind that erosive esophagitis is commonly found in 10 to 30% of LPR patients with an even lower proportion of patients (<10%) demonstrating Barrett’s metaplasia [74,97,98].

### 4.5. Pepsin Saliva Concentration

The sensitivity of pepsin saliva measurements ranges from 29.4% to 86.6% according to the diagnostic pepsin concentration cutoff, time of saliva collection, number of samples, and the method of measurement [99,100]. Several investigations showed inconsistencies in the associations between HEMII-pH, pepsin saliva concentrations, and symptom and finding severities, which is attributed to the instability of saliva pepsin concentrations over time, and the potential contribution of other gastroduodenal enzymes in mucosa inflammation and the development of symptoms [1,2]. Nowadays, pepsin tests may be an adjunctive tool for some conditions associated with LPR, including primary burning mouth syndrome [101] or idiopathic dental disorder [102,103,104], but that requires future clinical and basic science studies.

The key diagnostic features can be summarized as follows:-The HEMII-pH tracing may specify the profile of LPR disease regarding pH, composition, and time of occurrence of pharyngeal reflux events, which may orientate the personalized treatment.-High-resolution manometry may be advised in patients with a history of esophageal motility disorder or those with therapeutic resistance.-GI endoscopy is recommended for patients with a history of GERD complications, patients resistant to treatment, or elderly individuals.-Pepsin tests are sensitive but not specific and could be considered in oral diseases associated with LPR.

## 5. Treatment

The personalized treatment of LPR needs to consider the patient’s clinical features (age, body mass index, history), lifestyle (job, anxiety, stress), diet, and current or previous medications [84]. The treatment may consider (i) the management of LPR etiology (autonomic nerve dysfunction and diet), (ii) the prescription of medications to treat the consequences of reflux (symptoms and associated comorbidities), and (iii) the posttreatment management of disease to control LPR symptoms over the long term, avoiding medication as much as possible.

### 5.1. Proton Pump Inhibitors

PPIs have been considered as the primary medical treatment of LPR for a long time. However, contrary to GERD, the PPI efficacy over the placebo was never demonstrated in LPR disease [10]. The poor efficacy of PPIs may be attributed to the weakly acidic or alkaline pH of most pharyngeal reflux events, and the lack of GERD-related symptoms that are associated with distal esophageal acid reflux [10,68]. Interestingly, Pizzorni et al. recently demonstrated the non-inferiority of alginate over PPIs in a randomized controlled trial based on a 2-month empirical therapeutic trial [105]. Nowadays, for selected patients with a high-risk of acid GERD or LPR (e.g., obese or overweight patients), PPIs may be considered in empirical therapeutic trials in combination with alginate or magaldrate, which both act on weakly acidic or nonacidic reflux events [1,9,68]. The personalized therapeutic approach using PPIs needs to consider the patient’s age for the drug selection. Indeed, the several PPI classes report different clearance properties, which is important in elderly patients [106,107,108,109,110]. Esomeprazole has a more rapid onset of action and less variation in clearance rates than omeprazole. It has been suggested that the drug clearance decreases with age, exaggerating some of the differences between the PPIs and increasing the risk of drug interactions [106]. The reduction in plasma clearance mainly concerns rabeprazole, pantoprazole, and lansoprazole and may increase by 50 to 100% [106,107,108,109]. Esomeprazole may be primarily used in elderly patients because its clearance is not significantly affected by age [109]. Note that elderly patients are commonly taking several medications and there may have some drug interaction risks between PPIs and some medications, e.g., antiretroviral (HIV) drugs, anti-HCV drugs, cytostatics (e.g., methotrexate, dasatinib, erlotinib, nilotinib), itraconazole, immunosuppressants, and clopidogrel [106,108]. Finally, literature reviews report that PPIs were mostly used twice daily in LPR patients [10,68] but in practice, this regimen was only supported by one clinical study [109]. Nowadays, there is little evidence supporting the use of twice daily PPIs in place of a once daily dose (morning, fasting).

Note that H2-histamine blocker use was not discussed in the present paper because they are less effective in terms of healing rates and symptom relief for GERD, esophagitis, and LPR [106].

### 5.2. Alginate and Magaldrate

A comprehensive empirical therapeutic approach of LPR should account for both acidic and non-acidic reflux, and should include the possibility of reactivation of tissue-bound pepsin within the laryngopharynx [9]. Alginate and magaldrate coalesce in the acidic environment of the stomach into a floating raft, which may last 1 to 4 h and physically prevents the refluxate from leaving the stomach. Sodium alginate may be endowed with bio-adhesive potential, a property primarily due to its polymer chain length and ionizable groups, that provides a protective biofilm on the mucosa of the esophagus and upper aerodigestive tract [8,111,112,113]. Unlike PPIs, alginate and magaldrate reduce the number of esophageal and pharyngeal reflux events and the deposition of enzymes in the upper aerodigestive tract mucosa [112,113]. Alginate may be effective in LPR when used alone [105] or when used in combination with PPIs [76]. In patients with HEMII-pH findings, alginate and magaldrate may be used when the pharyngeal reflux events occur, mainly post-meal or when patients with both LPR and GERD feel esophageal symptoms [80,84]. The clinical effectiveness of alginate and magaldrate was never compared in LPR disease. From a theoretical standpoint, magaldrate, which is a complex compound formed from aluminum hydroxide and magnesium hydroxide, could be more appropriate for patients with alkaline pharyngeal reflux events containing bile acids because magaldrate may directly bind bile acids and, consequently, decrease the damage to the mucosa [80,84].

### 5.3. Surgery

Fundoplicature is an effective treatment for GERD by reducing the backflow of gastric content into the esophagus and GERD symptoms. However, due to the gaseous nature of pharyngeal reflux events and the low proportion of GERD-related symptoms, the effectiveness of fundoplicature was never demonstrated in LPR disease [114]. A systematic review of surgical approaches for LPR reported an important heterogeneity in the inclusion criteria of patients, with some GI surgeons proposing surgery to patients with LPR and GERD, whereas others included all LPR patients without consideration of GERD [114]. According to the lack of evidence about the usefulness of fundoplicature and the lack of investigation on the predictive value of HEMII-pH tracing on the surgery’s efficacy, the indication of this approach may be considered only in patients with troublesome or recalcitrant GERD symptoms or findings, e.g., esophagitis and Barrett’s disease. In the case of surgery, it remains important to explain to the patient that the surgery’s effectiveness on LPR symptoms remains unpredictable.

### 5.4. Long-Term Management

Due to the cost burden of LPR treatment for Western countries [78], long-term management is an important issue for healthcare systems. According to studies, the weaning rates of antireflux therapies in LPR patients ranged from 64% to 75% of cases [115,116,117], making the long-term prescription of PPIs or other drugs controversial. Indeed, more than 50% of LPR are acute or recurrent [51], and, consequently, do not require long-term medication. The long-term management of LPR needs to consider the patient features and may act on their diet and lifestyle (stress, anxiety), while considering patient histories, comorbidities, and wishes. Indeed, the adherence to an antireflux diet may be difficult for patients with refluxogenic diet sensitivity, and some of them will prefer taking long-term drugs. In this case, the use of medications with the lowest risk of adverse events makes sense, such as alginate or magaldrate. The long-term use of PPIs has long been suspected to be associated with adverse events, including pneumonia, acute coronary syndromes, colitis, altered mineral and vitamin absorption, and fractures and related orthopedic injury, but the evidence level of these associations is B or C [118,119,120]. Most patients on long-term PPI therapy and practitioners may be unaware about these potential adverse events [115,118]. A personalized management algorithm considering patient and reflux features is proposed in Figure 5. It is important to note that implementing personalized approaches relies on extensive patient data and ensuring the privacy and security of sensitive information. Consequently, personalized approaches require future ethical considerations for data protection, especially when an artificial intelligence software is used to collect medical information. Because developing and implementing personalized therapeutic strategies may be resource-intensive, requiring advanced diagnostic tools and technologies, the current resource limitations of many healthcare systems may be considered the primary barrier to the widespread adoption of personalized approaches. The development of artificial intelligence software supporting the physician’s tasks may be an important issue for widespread future personalized approaches [121,122,123,124,125].

The key treatment features can be summarized as follows:-The management of LPR etiologies, e.g., diet and lifestyle, is the primary therapeutic step, while the use of medication may just control the LPR consequences, such as symptoms and the associated conditions.-The medical and surgical histories, lifestyle, diet, and medications of patients, especially the elderly, need to be considered in the selection of drugs, especially PPIs.-In patients with HEMII-pH findings, the medical treatment may be personalized according to the features of the esophageal and pharyngeal reflux events through a combination of PPIs and alginate or magaldrate.-In patients without HEMII-pH findings, the personalized approach may be focused on patient characteristics rather than the HEMII-pH tracing and may consider alginate/magaldrate with or without PPIs.-Fundoplicature may be proposed in patients with troublesome or recalcitrant GERD symptoms or complications, although the surgery effectiveness for LPR symptoms and findings remains unpredictable.-Most patients may be weaned from all medications but the etiological factors of LPR need to be controlled over the long-term, which may be difficult for patients with chronic autonomic nerve dysfunction or a high sensitivity to a refluxogenic diet.

## 6. Conclusions and Future Directions

Laryngopharyngeal reflux is a prevalent condition in Western populations, which has been ignored for a long time because of the definition, diagnosis, and therapeutic controversies. However, the disease has gained in popularity in the past decade regarding the scientific advances and the better identification of factors that may influence the clinical presentation and therapeutic management. Accordingly, the consideration of more personalized management approaches makes sense and should include patient age, weight, gender, history, and reflux features. Future studies are needed to better understand the various patient factors, e.g., demographics, lifestyle, and comorbidities, for different populations to study and improve the external validity of personalized approaches.

## Figures and Tables

**Figure 1 jpm-13-01567-f001:**
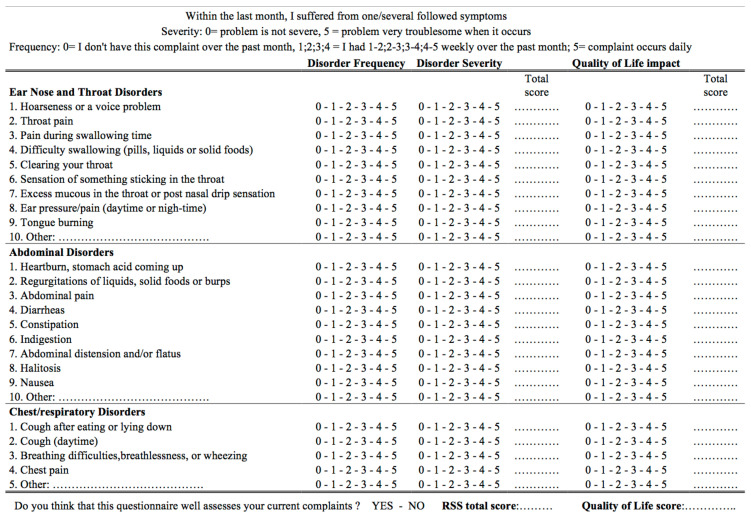
Reflux symptom score. The questionnaire is subdivided into three parts according to the complaints: ear, nose, and throat (part 1, 9 items); digestive (part 2, 9 items); and respiratory (part 3, 4 items) symptoms. The frequency and severity of each symptom are rated on a 5-point scale. Regarding the frequency, 0 = patient did not have the complaint over the past month; 1, 2, 3, 4 = patient had the complaint 1–2, 2–3, 3–4, 4–5 times weekly over the past month; 5 = patient had the complaint daily over the past month. Regarding the severity, 0 = the complaint was absent and 5 = the complaint was very troublesome when it occurs. For each item, the severity score is multiplied by the frequency score to obtain a symptom score ranging from 0 to 25. The sum of these symptom scores is calculated to obtain the final RSS score (ranging from 0 to 550; with the possibility for the physician and the patient to add 3 symptoms not identified in the RSS, leading to a maximum possible score of 625). The RSS also assesses the symptoms’ impact on quality of life. The total quality of life score is calculated as the sum of each item score. For example, a patient who reports a very mild dysphonia voice problem every day of the week will have a score of item 1 of 5 × 1 = 5, while the impact on quality of life will range from 0 (no impact) to 5 (severe impact). In the case of very severe daily nausea, the score of item “nausea” will be 5 × 5 = 25, with a QoL item score of 5.

**Figure 2 jpm-13-01567-f002:**
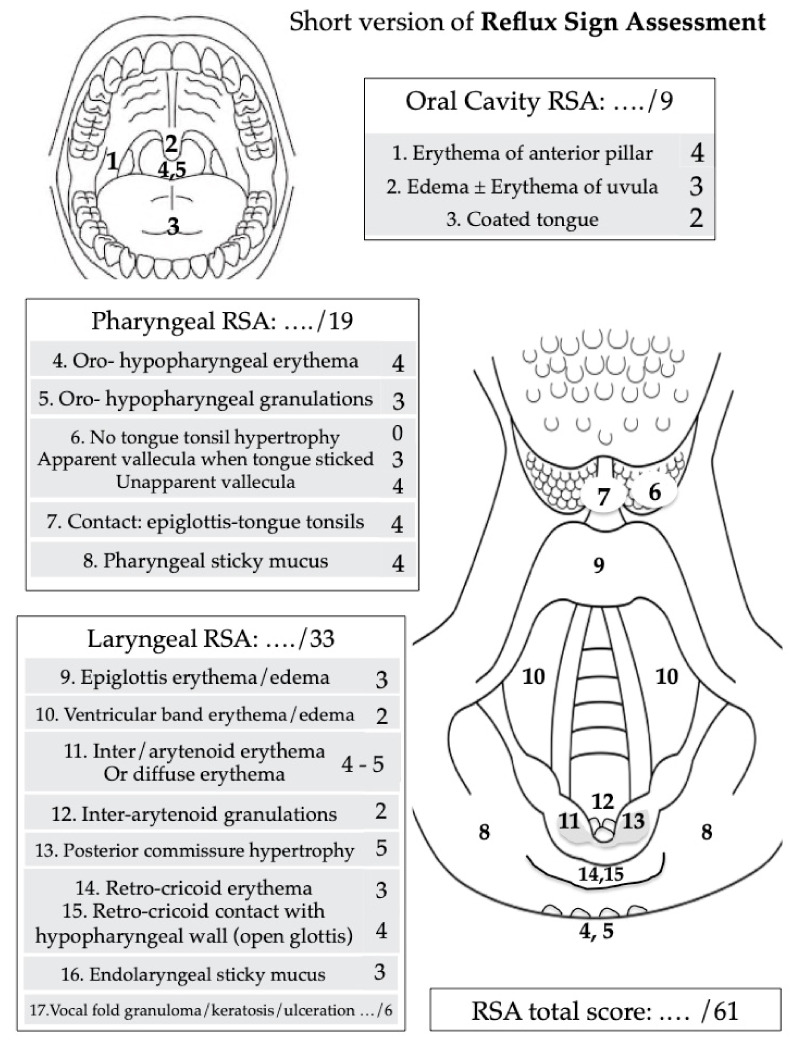
Reflux sign assessment. The tool is subdivided into three parts according to the sign localization: oral cavity, pharynx, and larynx. The occurrence of vocal fold granuloma (+2), keratosis (+2), or ulceration (+2) may be considered in the last item of the score. Because of the low prevalence, the following items were removed from the initial version of RSA (in the RSA validation paper): edema/erythema of the vocal folds, nasopharyngeal erythema, and subglottic edema/erythema. The total score is calculated as the sum of each item score. The maximum score is 61.

**Figure 3 jpm-13-01567-f003:**
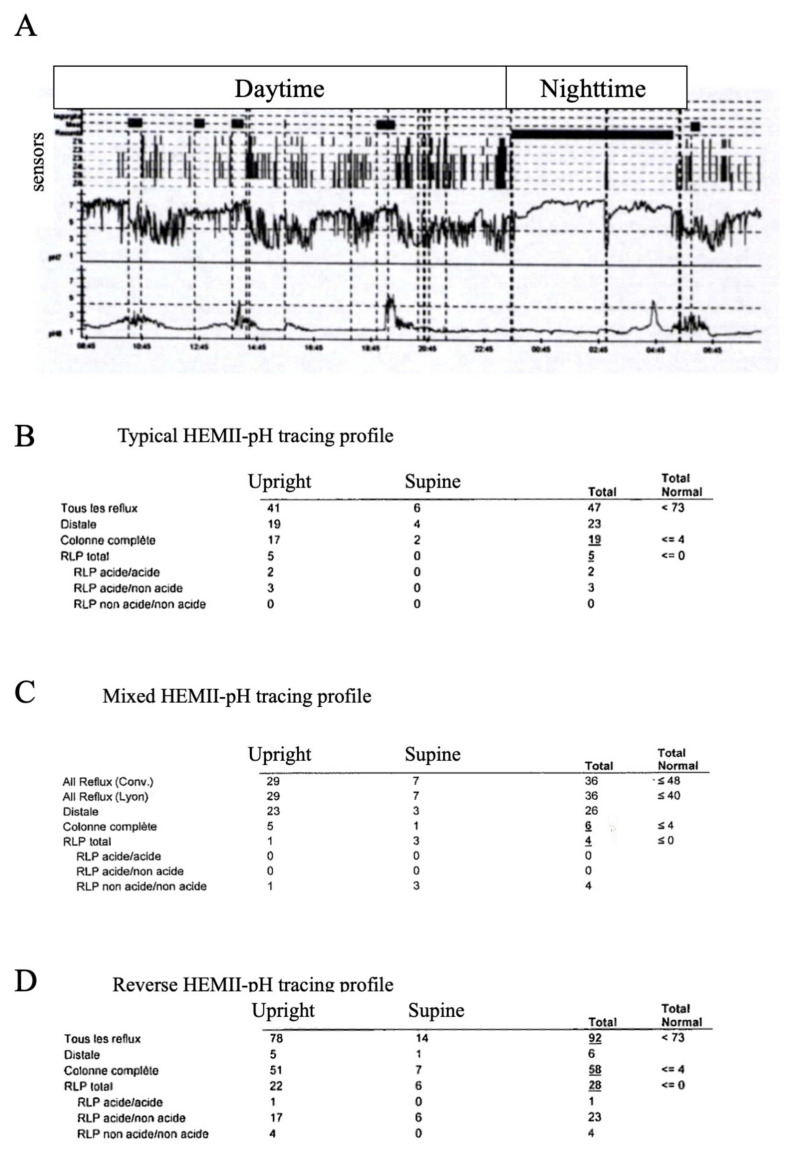
pH-impedance tracing profiles. Three main profiles of LPR patients at the HEMII-pH may be observed: patients with only daytime and upright hypopharyngeal reflux episodes (**A**,**B**); patients with a mixed profile including daytime/upright and nighttime/supine pharyngeal episodes (**C**); and patients with a reverse tracing consisting of supine and upright pharyngeal reflux events (**D**). Abbreviations: HEMII-pH = hypopharyngeal esophageal multichannel intraluminal impedance-pH monitoring; LPR = laryngopharyngeal reflux.

**Figure 4 jpm-13-01567-f004:**
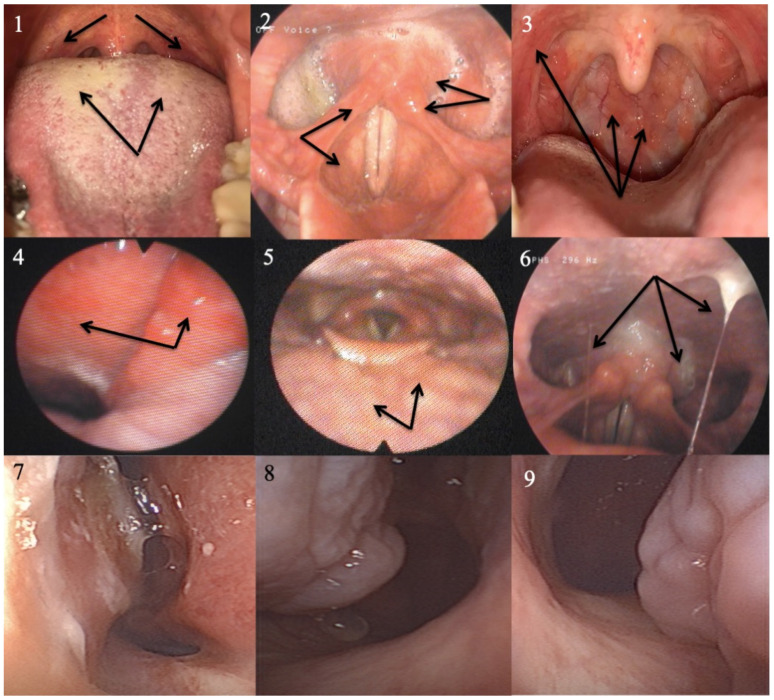
Typical and atypical signs associated with laryngopharyngeal reflux. Laryngopharyngeal reflux may present with laryngeal, pharyngeal, oral, and nasal signs, including coated tongue (**1**), anterior pilar erythema (**1**), laryngeal erythema (**2**), heterogeneous erythema of the posterior oropharyngeal wall (**3**), nasopharyngeal erythema (**4**), tongue tonsil hypertrophy (**5**), sticky throat mucus (**6**), nasal mucosa dryness (**7**), and mulberry inferior turbinate (**8**,**9**).

**Figure 5 jpm-13-01567-f005:**
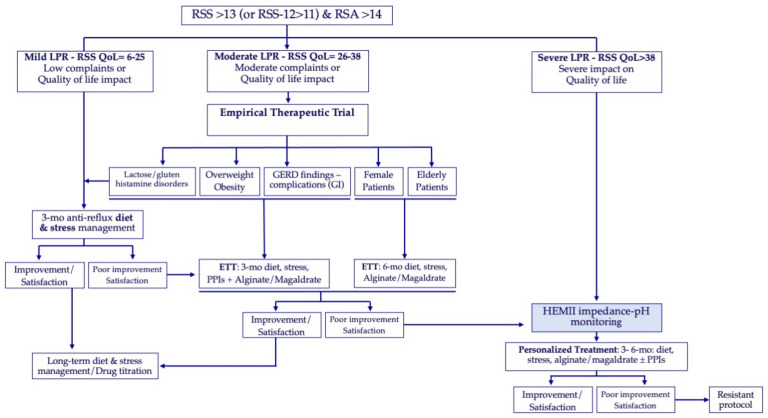
Personalized algorithm for reflux management. The RSS-QoL thresholds were determined through receiver operating characteristic [51]. Abbreviations: ETT = empirical therapeutic trial; HEMII-pH = hypopharyngeal–esophageal multichannel intraluminal impedance-pH monitoring; mo = months; PPI = proton pump inhibitor; QoL = quality of life; RSA = reflux sign assessment; RSS = reflux symptom score.

**Table 1 jpm-13-01567-t001:** Dubai criteria for diagnosis of laryngopharyngeal reflux using impedance-pH monitoring.

	Hypopharyngeal–Esophageal Multichannel Intraluminal Impedance pH-Monitoring Outcomes and Diagnosis Criteria
1	Single-channel (esophageal) or dual-channel (esophageal-esophageal) pH probes are useful for diagnosing GERD but not adequate for diagnosing LPR
	because of the lack of a pharyngeal sensor and lack of consideration of non-acidic events.
2	If HEMII-pH is unavailable, an empirical treatment covering acidic, weakly acidic, and nonacidic LPR may be prescribed and evaluated at 3 months.
	Treatment success of LPR should be based on improvement of the patient’s LPR symptoms.
3	The HEMII-pH results may provide guidance as to the appropriate nature, dosing, and timing of medications for the specific patient according to
	the type of LPR (acidic, weakly acidic, nonacidic) and time of occurrence (upright and daytime and/or nighttime)
4	Triple-channel (dual esophageal and pharyngeal) pH-only studies may detect acidic pharyngeal reflux events but they are not sufficient to rule out
	LPR disease as they may miss weakly acidic and nonacidic pharyngeal events.
5	HEMII-pH monitoring has to respect the following placement characteristics:
	(1) Proximal pH sensor in the hypopharyngeal cavity at 0.5 cm to 1 cm above upper esophageal sphincter or within the sphincter.
	(2) Distal pH sensor in the esophagus as close to 5 cm above lower esophageal sphincter as possible.
	(3) At least 2 impedance sensor pairs in the esophagus.
	(4) At least 1 impedance sensor pair in the pharyngeal cavity. It is recommended to control the placement of the upper pH sensor using flexible laryngoscopic
	or manometric guidance. The recommended duration of the examination is 24 h. During the 24 h testing, the patient should continue their normal
	diet and activities.
6	Based on HEMII-pH, a hypopharyngeal acidic event consists of an event with a pH < 4.0. A hypopharyngeal weakly acidic reflux event consists of
	an event with a pH between 4.0 and 7.0. A hypopharyngeal alkaline reflux event consists of an event with pH > 7.0.
7	The analysis of the 24 h recording must respect the following:
	(1) Exclusion of reflux events during meals;
	(2) Pharyngeal reflux event diagnosed only when a reflux event originating from the distal most impedance channel reaches the pharyngeal channels in
	a retrograde fashion;
	(3) Manual analysis to identify reflux events that the computer may have reported incorrectly.
8	The severity of LPR seen using HEMII-pH or oropharyngeal pH monitoring is not necessarily correlated with the severity of symptoms and findings.
9	While HEMII-pH is promising as an objective tool for diagnosing LPR, the correlation between its findings and treatment outcomes remains limited.
	Controlled studies are needed to validate the value of this technology in predicting treatment outcomes.
10	Reflux monitoring for LPR, whether using HEMII-pH, MII-pH, or pH measurements, should be performed off acid suppression medications, beginning
	at least 7 days prior to the study.
11	The LPR diagnosis may not be confirmed with esophageal catheters that are configured with two esophageal pH sensors and without impedance or
	pH sensors in the pharynx because (1) the proximal esophageal reflux events may not reach the hypopharynx and (2) the presence of reflux events near
	the UES may be altered by swallowing saliva.
12	Hypopharyngeal–esophageal multichannel intraluminal impedance pH monitoring (HEMII-pH) is an objective tool to identify esophago-pharyngeal
	reflux events (acidic, weakly acidic, or nonacidic) and can suggest a diagnosis of LPR when there is >1 hypopharyngeal reflux event in 24 h.

Abbreviations: HEMII-pH = hypopharyngeal–esophageal multichannel intraluminal impedance-pH monitoring; GERD = gastroesophageal reflux disease; LPR = laryngopharyngeal reflux disease.

## Data Availability

Not applicable.

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
