# Peer review of "Personalized Treatments Based on Laryngopharyngeal Reflux Patient Profiles: A Narrative Review"

_jpm, 2023, doi:10.3390/jpm13111567_

Round 1

Reviewer 1 Report

Comments and Suggestions for Authors

This review critically examines current literature on laryngopharyngeal reflux (LPR), aiming to identify distinct patient profiles and propose personalized diagnostic and therapeutic strategies. Employing PubMed, Scopus, and the Cochrane Library databases, a state-of-the-art review was conducted, extracting information on epidemiology, demographics, clinical presentations, diagnostic approaches, and therapeutic responses associated with LPR.

While the proposed approach of personalized diagnostic and therapeutic strategies for laryngopharyngeal reflux (LPR) has several merits, there are potential drawbacks and challenges to consider:

1.       Data Availability and Privacy Concerns: Implementing personalized approaches often relies on extensive patient data. Ensuring the privacy and security of this sensitive information is crucial. Ethical considerations and adherence to data protection regulations must be prioritized.

2.       Limited Data on Enzyme Profiles: The review acknowledges a lack of information about the role of other gastrointestinal enzymes in the development of LPR-related mucosa inflammation. The scarcity of data in this area may hinder the development of comprehensive enzyme-based personalized approaches.

3.       Complexity of Patient Profiles: Personalized medicine requires a nuanced understanding of various patient factors, including demographics, lifestyle, and comorbidities. Managing the complexity of diverse patient profiles may pose challenges in implementing standardized personalized strategies.

4.       Resource Intensiveness: Developing and implementing personalized therapeutic strategies may be resource-intensive, requiring advanced diagnostic tools and technologies. This could be a barrier to widespread adoption, particularly in healthcare systems with limited resources.

5.       Generalizability and External Validity: The effectiveness of personalized approaches may vary across populations. Strategies developed based on a specific set of patients might not be easily generalized to diverse populations, limiting the external validity of such approaches.

Author Response

1.Data Availability and Privacy Concerns: Implementing personalized approaches often relies on extensive patient data. Ensuring the privacy and security of this sensitive information is crucial. Ethical considerations and adherence to data protection regulations must be prioritized.

We add this sentence p. 17, line 392: It is important to note that implementing personalized approaches relies on extensive patient data, ensuring the privacy and security of sensitive information. Consequently, personalized approaches require future ethical considerations for data protection, especially when an artificial intelligence software is used to collect medical information.”

2.Limited Data on Enzyme Profiles: The review acknowledges a lack of information about the role of other gastrointestinal enzymes in the development of LPR-related mucosa inflammation. The scarcity of data in this area may hinder the development of comprehensive enzyme-based personalized approaches.

We add the following sentence: p.2, line 67: “However, a personnalized approach based on enzyme profile requires future studies to determine the role of all enzymes in the development of mucosa lesions and inflammation.”

  1. Complexity of Patient Profiles: Personalized medicine requires a nuanced understanding of various patient factors, including demographics, lifestyle, and comorbidities. Managing the complexity of diverse patient profiles may pose challenges in implementing standardized personalized strategies.

We agree. We add in the conclusion of the paper this information: p.19, line 420: “Future studies are needed to better understand various patient factors, e.g. demographics, lifestyle, and comorbidities.”

  1. Resource Intensiveness: Developing and implementing personalized therapeutic strategies may be resource-intensive, requiring advanced diagnostic tools and technologies. This could be a barrier to widespread adoption, particularly in healthcare systems with limited resources.

We add in the paper: p.17, line 396: “Because developing and implementing personalized therapeutic strategies may be resource-intensive, requiring advanced diagnostic tools and technologies, the current resource limitations of many healthcare systems may be considered as the primary barrier to widespread adoption of personalized approaches. The development of artificial intelligence software supporting the physician tasks may be an important issue to widespread future personalized approaches.”

  1. Generalizability and External Validity: The effectiveness of personalized approaches may vary across populations. Strategies developed based on a specific set of patients might not be easily generalized to diverse populations, limiting the external validity of such approaches.

We add in the conclusion: p.19, line 424: “Future studies are needed to better understand various patient factors, e.g. demographics, lifestyle, and comorbidities per populations to study and improve the external validity of personalized approaches.”

Reviewer 2 Report

Comments and Suggestions for Authors

In the review by Dr. Lechien talks about a thorough look into existing research on laryngopharyngeal reflux (LPR), which is a condition affecting the throat due to stomach and fluid reflux. The authors explored various studies using well-known research databases to understand different aspects of LPR, such as who it affects, how it presents itself, and how it’s currently diagnosed and treated. Factors like age, gender, weight, other health conditions, eating habits, cultural aspects, mental stress, and certain enzymes in saliva appear to impact how LPR shows up and responds to treatment. A detailed understanding of reflux symptoms (which can affect the throat, respiratory system, nose, and eyes) and specialized testing is critical for managing patient care effectively. Although personalized treatment approaches have shown promising results in relieving symptoms and providing long-term cures, the lack of information about the role of certain stomach enzymes in LPR limits developing new, enzyme-focused treatment and diagnostic strategies. The author concludes by highlighting the difficulty in managing LPR, and suggests using artificial intelligence to create more customized treatment plans based on individual patient characteristics. The manuscript overall in well written logic and list the sufficient publications to support. I have some minor comments for the author’s consideration.

(1)   In figure 5, the author listed the QoL (quality of life) score cutoff to evaluate the severity of disease and the corresponding treatment. Could the author provide an explanation of how to decide these cutoff numbers? Are these numbers based on meta-analysis, logistic regression, or ROC analysis?

(2)   For the figure 1, could author provide an example of disease score calculation to enable the reader to have a better understanding?

(3)   In line 25-26, the author proposed the leverage of AI for precise treatment. However, the main manuscript does not elaborate enough on the advantage of AI. Could the author list one example, (even in another disease field) that a large patient training dataset that enables the AI to reason and learn, and empowers clinicians to precisely identify the phenotype of patients, and provide precise medicine? 

Comments on the Quality of English Language

none

Author Response

1.In figure 5, the author listed the QoL (quality of life) score cutoff to evaluate the severity of disease and the corresponding treatment. Could the author provide an explanation of how to decide these cutoff numbers? Are these numbers based on meta-analysis, logistic regression, or ROC analysis? 

These thresholds were determined in the following publication: Lechien JR, Lisan Q, Eckley CA, Hamdan AL, Eun YG, Hans S, Saussez S, Akst LM, Carroll TL. Acute, Recurrent, and Chronic Laryngopharyngeal Reflux: The IFOS Classification. Laryngoscope. 2023; 133(5):1073-1080. doi: 10.1002/lary.30322.

We add in the Figure 5 footnotes: “The RSS-QoL thresholds were determined through receiver operating characteristic [51].”

(2)   For the figure 1, could author provide an example of disease score calculation to enable the reader to have a better understanding?

We add in the figure footnotes: “For example, a patient who reports dysphonia a very mild voice problem every day of the week will have a score of item 1 at 5x1=5; while the impact on quality of life will range from 0 (no impact) to 5 (severe impact). In case of very severe daily nausea, the score of item “nausea” will be 5x5=25, with a QoL item score of 5.

(3)   In line 25-26, the author proposed the leverage of AI for precise treatment. However, the main manuscript does not elaborate enough on the advantage of AI. Could the author list one example, (even in another disease field) that a large patient training dataset that enables the AI to reason and learn, and empowers clinicians to precisely identify the phenotype of patients, and provide precise medicine? 

We add in the paper: p.17, line 396: “Because developing and implementing personalized therapeutic strategies may be resource-intensive, requiring advanced diagnostic tools and technologies, the current resource limitations of many healthcare systems may be considered as the primary barrier to widespread adoption of personalized approaches. The development of artificial intelligence software supporting the physician tasks may be an important issue to widespread future personalized approaches.”